# A Practical Approach to Gear Design and Lubrication: A Review

**Dario Croccolo** [1] , **Massimiliano De Agostinis** [1], **Giorgio Olmi** [1] **and Nicolò Vincenzi** [2,*]

[1] Department of Industrial Engineering (DIN), University of Bologna, Viale del Risorgimento, 2, 40136 Bologna, Italy; dario.croccolo@unibo.it (D.C.); m.deagostinis@unibo.it (M.D.A.); giorgio.olmi@unibo.it (G.O.)

[2] GIULIANI, A Bucci Automations S.p.A. Division, Via Granarolo 167, 48018 Faenza, Italy

\* Correspondence: n.vincenzi@bucci-industries.com

**Abstract:** The modern design of mechanical parts, such as gears, goes through the continuous demand for a high level of efficiency and reliability, as well as an increased load carrying capacity and endurance life. The aim of the present paper was to perform a review and to collect practical examples in order to provide interesting tips and guidelines for gear design, including both its dimensioning and its lubrication. From this point of view, this paper is particularly novel, as it is a full-comprehensive collection of all the tools supporting gear design. Several practical aspects have been taken into account, including the definition of the right profile shifting, the selection of a proper lubricant, and the definition of the quality grade and of the tolerances needed to obtain the correct backlash. Finally, a numerical example is provided, addressing the research of the best solution to fit a given space, while maximizing the transmittable torque over weight ratio for two mating spur gears.

**Keywords:** gear; profile shift; gear optimization; gear tolerances; gear lubrication

## 1. Introduction

The modern design of mechanical parts, such as gears, goes through the continuous demand for a high level of efficiency and reliability, as well as an increased load carrying capacity and endurance life. At the same time, the concepts of "robust design" and "optimization" involve nowadays the need for a very high-power density (involving smaller size, lower weight, lower noise and vibrations), as well as longer service intervals [1,2]. Finally, the aspects related to sustainability (analysis of the impact on the environment) and low-cost design are also commonly regarded as driving forces in the development of modern products. Within this framework, gear design must address, for example, low consumption of lubrication oil, keeping hydraulic losses as low as possible [3]. Oil/air mist (micro-fog) may be used for bearing and gear lubrication, especially in combination with compact design approaches, for instance in the field of machine tool industry. Regarding this point, an applicative example is shown in Figure 1. A continuous lubrication can be achieved by this strategy, involving very low oil quantities with consequent consumption sharp reduction. Depending on the application, the type of oil, whose main feature is viscosity, can be selected between ISO VG32 (high speed) and ISO VG 68 (high load). The oil flow, $Q$ [mm$^3$/h], being needed for lubrication by micro-fog, can be evaluated by Equation (1), where $D_m$ is the mean bearing diameter or the gear pitch diameter.

$$Q = 1.2 \cdot D_m \ to \ 1.3 \cdot D_m \tag{1}$$

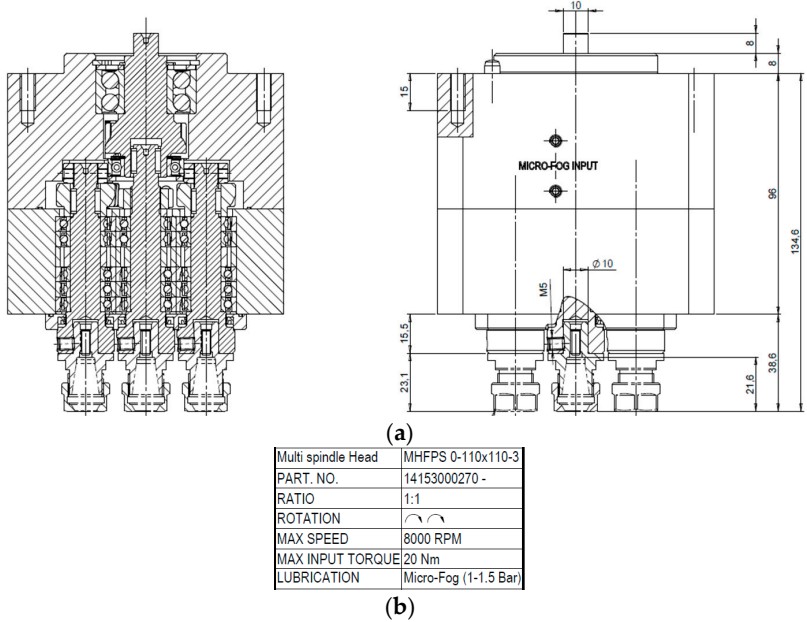

**Figure 1.** Example of multi-spindle head: gear and bearings lubricated by micro-fog: (**a**) drawing and (**b**) table.

With reference to the case study in Figure 1, a MOBIL DTE LIGHT 32 oil and a centralized unit, having the capability of providing (by a suitable amount of nozzles) 5 to 30 mm$^3$/cycle, were selected. The required oil flow, $Q$ [mm$^3$/h], depending on the number and size of bearings and gears, was estimated as $Q$ = 3000 mm$^3$/h. The nozzles were set at 15 mm$^3$/cycle; therefore, the number of cycles requested to the nozzles, to provide the required flow was set to 200 cycles per hour. The air pressure may be set for this application between 0.3 and 2.0 bar, depending on the dimension of the components (the bigger components, the lower the pressure level).

As for lubrication, the selection of the material and its thermal treatment is the other key choice, to increase the power density: case hardened and nitrided gears are the two most common alternative options [4–8]. Both treatments warrant improved contact (and bending) fatigue performance. The main differences between the two treatments, from the point of view of their application, are a consequence of the manufacturing cycle and of the distortions arising from the thermo-chemical process. The high temperature of the carburizing process induces larger distortions that entail subsequent machining, being usually conducted by grinding, and the consequent amount of allowance to be removed. The lower temperature of nitriding treatments allows for finishing process (milling or grinding) being completed before heat treatment, taking advantage of lower distortions. However, nitriding has the drawback of generally requiring a longer treatment duration. Moreover, it produces lower case depths, according to Figure 2.

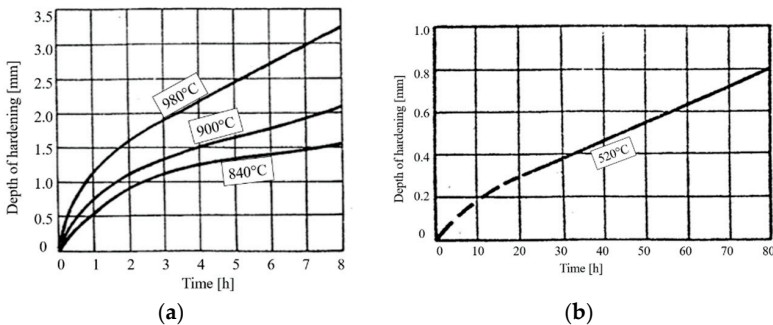

**Figure 2.** Depth of hardening vs time according to Bucci Automations internal standards: (**a**) case hardened treatment on steel 18NiCrMo5; (**b**) nitriding treatment on steel 41CrAlMo7.

In the past 20 years, gear design has been approached from these two points of view:

- increasing power density as much as possible, which has led to gears with asymmetric teeth, according to Reference [9,10]; and
- speeding-up gear production time, which has led to additively manufactured gears [11].

As for asymmetric gears, today, a modern 5-axes Computerized Numerical Control (CNC) machine makes it possible to cut a tooth, where the drive side is differently shaped, if compared to the coast side (see Figure 3). The effect of this tooth profile alteration can be twofold:

- Keeping the same drive side pressure angle (i.e., 20°), while increasing the coast side pressure angle (i.e., 25° to 35°), it is possible to increase tooth bending strength or to obtain the same strength for a thinner tooth head. The benefit arising from this improved shape can be expressed as a function of the difference between the drive and coast pressure angles, according to reference [12]. However, this design does not affect the actual load capacity that is limited by high contact stresses, as no change in drive side pressure angle occurs.
- Increasing drive side pressure angle, according to Reference [9], it is possible to drop down the stresses (due to both bending and contact) and, furthermore, to also reduce the vibrational level.

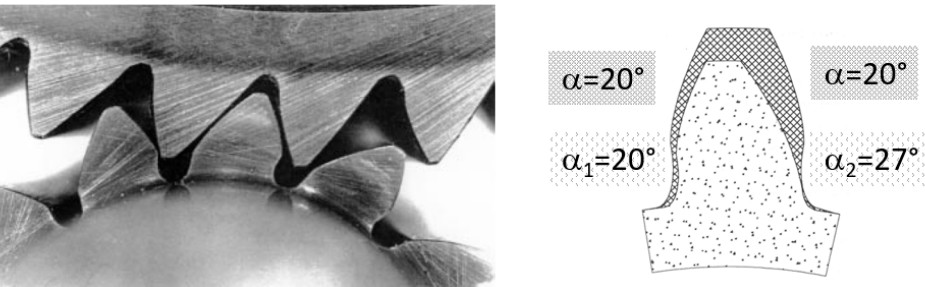

**Figure 3.** Example of asymmetric tooth.

With regard to additively manufactured gears, a recent study [11] highlighted the performance of gears produced by Selective Laser Melting: the retrieved mechanical strength is generally lower than that of wrought material gears, especially under fatigue [13], mainly due to the induced porosities and residual stresses, but the advantage in terms of saved production time is not negligible. In addition, it is worth mentioning that forged bi-metal gears can be fabricated today [14,15].

This review paper aimed at steering the designer through gear development, based on industrial specifications, including, on one hand, its design, dimensioning, and final drawing, and, on the other hand, its most suitable lubrication. Regarding the first point, this paper deals with a collection of formulas, tables and graphs taken from standards and scientific and technical studies in the literature. These formulas have sometimes been revisited, to account for gear design (instead of just verification) purposes. As for the second point, the guidelines for lubrication this paper provides are based on the available tools in the technical literature and are strictly related to the main features of the industrial application to be developed. Issues of novelty arise from the lack of scientific or technical review studies, addressing this topic and providing what can be regarded as a full-comprehensive collection of the aforementioned tools that are expected to significantly support the design task.

## 2. Practical Design of Gears

The structural assessment of mating gears can be carried out by well-known standards, such as ISO [16] or American Gear Manufacturers Association, AGMA [17], thus assessing gear load capacity against both bending stresses and pitting (contact) stresses. The verification is based on the calculation of several factors that affect the nominal stresses, as well as the allowable stresses. The stress analysis may also be deepened further determining local stresses for fatigue life prediction, according to

Reference [18–21]. However, these procedures require the full definition of gear geometry: In other words, verification and accurate structural assessment are possible, but gear design, starting from the white paper, is almost impossible, unless an iterative methodology is exploited. In particular, a simplified (practical) procedure must be applied, in order to coarsely design the transmission, thus defining all the starting parameters to allow for verification. This practical procedure must also account for the possible need for profile shifting (both with or without center distance variation): For this purpose, optimal values for the profile shifting coefficient should be made available to the designer. The approach in the present paper is based on the Nomenclature in the Appendix A, also according to Figure 4:

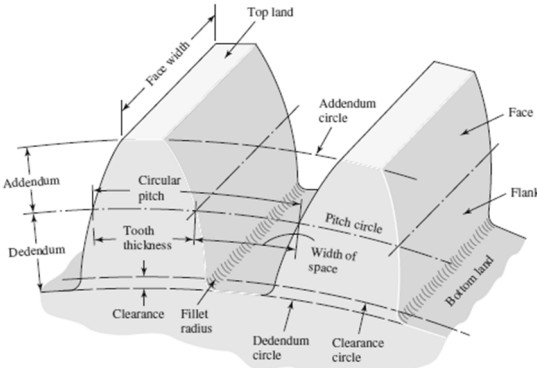

**Figure 4.** Gear tooth Geometry, see Nomenclature (taken from Reference [22]).

In order to warrant motion regularity, the (transverse) contact ratio must be $\varepsilon_\alpha >$ [1.25 to 1.4].

An example of calculation for $\varepsilon_\alpha$ is reported in Table 1.

For powertrain design, the initially available data, to be regarded as specifications, are usually the input/output torque $M_t$ [Nm], the input/output rotational speed $n$ [rpm] (and therefore the transmission ratio $\tau$), and the space available for the application, namely the theoretical wheelbase I [mm]. It is first suggested to evaluate (in a very fast and approximated way) a starting module $m$, based on the following Equation (2):

$$m \geq Y \cdot \sqrt[3]{\frac{M_t \cdot 1000}{\lambda \cdot \sigma_{amm}}}. \tag{2}$$

In Equation (2), $\lambda$ can be chosen by the designer in the following ranges (practical value $\lambda$= 8 to 12):

- $6 < \lambda < 10$ for gearbox applications (characterized by the need for gear shifting) or for gears mounted on cantilever shaft;
- $15 < \lambda < 25$ for gears on rigid supports;
- $25 < \lambda < 45$ for gears on very rigid supports with rigid shaft; and Y (derived from the Lewis coefficient) can be considered equal to:
- 0.8 for $z < 20$;
- 0.6 for $20 < z < 40$;
- 0.4 for $z > 40$.

**Table 1.** Calculated values of contact ratio $\varepsilon_\alpha$ for spur gear using standard tool of 20° pressure angle according to ISO 53 [23] with no addendum modification (no profile shift).

| $I_i$ = 100 mm | | | | $I_i$ = 200 mm | | | | $I_i$ = 500 mm | | | |
|---|---|---|---|---|---|---|---|---|---|---|---|
| $z_2/z_1$ | $z_1$ | $m$ | $\varepsilon_a$ | $z_2/z_1$ | $z_1$ | $m$ | $\varepsilon_a$ | $z_2/z_1$ | $z_1$ | $m$ | $\varepsilon_a$ |
| 1 | 100 | 1 | 1.853 | 1 | 200 | 1 | 1.912 | 1 | 125 | 4 | 1.875 |
| | 50 | 2 | 1.755 | | 160 | 1.25 | 1.896 | | 100 | 5 | 1.853 |
| | 25 | 4 | 1.612 | | 100 | 2 | 1.853 | | 50 | 10 | 1.755 |
| | 20 | 5 | 1.557 | | 80 | 2.5 | 1.826 | 1.5 | 160 | 2.5 | 1.909 |
| 1.5 | 80 | 1 | 1.849 | | 50 | 4 | 1.755 | | 100 | 4 | 1.872 |
| | 64 | 1.25 | 1.821 | | 40 | 5 | 1.714 | | 80 | 5 | 1.849 |
| | 40 | 2 | 1.749 | | 25 | 8 | 1.612 | | 50 | 8 | 1.786 |
| | 32 | 2.5 | 1.708 | | 20 | 10 | 1.557 | | 40 | 10 | 1.749 |
| | 20 | 4 | 1.606 | 1.5 | 160 | 1 | 1.909 | 3 | 125 | 2 | 1.909 |
| | 16 | 5 | 1.551 | | 128 | 1.25 | 1.893 | | 100 | 2.5 | 1.893 |
| 3 | 50 | 1 | 1.823 | | 80 | 2 | 1.849 | | 50 | 55 | 1.823 |
| | 40 | 1.25 | 1.792 | | 64 | 2.5 | 1.822 | | 25 | 10 | 1.714 |
| | 25 | 2 | 1.714 | | 40 | 4 | 1.749 | 4 | 200 | 1 | 1.937 |
| | 20 | 2.5 | 1.671 | | 32 | 5 | 1.708 | | 160 | 1.25 | 1.927 |
| 4 | 40 | 1 | 1.805 | | 20 | 8 | 1.605 | | 100 | 2 | 1.899 |
| | 32 | 1.25 | 1.773 | | 16 | 10 | 1.550 | | 80 | 2.5 | 1.881 |
| | 20 | 2 | 1.691 | 3 | 100 | 1 | 1.893 | | 50 | 4 | 1.833 |
| | 16 | 2.5 | 1.647 | | 50 | 2 | 1.823 | | 40 | 5 | 1.805 |
| | | | | | 25 | 4 | 1.714 | | 25 | 8 | 1.732 |
| | | | | | 20 | 5 | 1.671 | | 20 | 10 | 1.691 |
| | | | | | 10 | 10 | 1.511 | | | | |
| | | | | 4 | 80 | 1 | 1.881 | | | | |
| | | | | | 64 | 1.25 | 1.860 | | | | |
| | | | | | 40 | 2 | 1.805 | | | | |
| | | | | | 32 | 2.5 | 1.773 | | | | |
| | | | | | 20 | 4 | 1.691 | | | | |
| | | | | | 16 | 5 | 1.647 | | | | |

The allowable stress can be assumed as $\sigma_{all} \sim 150$ MPa in the case of gear made of steel. The designer is then expected to select the first available module, according to the ones in ISO 54 [24]. Subsequently, based on the starting module, it is possible to address gear design, following the steps below (input data: *I*, $\tau$).

The number of teeth $z_{1,2}$ can be selected as (Equation (3)):

$$z_1 = \frac{2 \cdot I}{m \cdot \left(1 + \frac{1}{\tau}\right)}; \; z_2 = \frac{z_1}{\tau}. \tag{3}$$

The tooth can be designed according to modular ISO 53 [23] proportion ($a = m$; $d = 1.25\,m$) or to stub proportion ($a = 0.8\,m$; $d = m$). Stub proportioning is suitable for highly stressed gears (generally operating under high torque, low speed) because the tooth is made stiffer. On the other hand, it must be remarked that stub design is affected by a reduced contact ratio with respect to the

modular proportioning. The designer has to then calculate the contact ratio $\varepsilon_\alpha$ and verify the condition for motion regularity ($\varepsilon_\alpha > [1.25$ to $1.4]$), as well as the further condition regarding the minimum number of teeth threshold, to avoid under-cut (Equation (4)) without profile shifting:

$$z_{1\_min} = \frac{2 \cdot a}{m \cdot sen^2 \vartheta}; \ z_{1\_min\_ISO} = 18; \ z_{1\_min\_STUB} = 14 \tag{4}$$

The following step is estimating the needed profile shifting [25]. This strategy (generally regarded as positive profile shifting) can be used to obtain different advantages, such as:

- avoiding under-cut;
- improving the response of the pinion gear against bending and pitting; and
- modifying the wheelbase (working wheelbase different from theoretical wheelbase).

The working wheelbase $I_i$ [mm] is usually selected by the designer and, as further steps, the following items may be calculated: (i) the working pressure angle $\vartheta_i$ [°]; (ii) the addition of profile shifting $x_1 + x_2$ [mm]; or (iii) their dimensionless coefficients $\delta_1 + \delta_2$ by Equation (5):

$$\begin{cases} \cos(\vartheta_i) = \frac{I \cdot \cos(\vartheta)}{I_i} \\ x_1 + x_2 = \frac{z_1 + z_2}{2 \cdot \tan(\vartheta)} \cdot (inv(\vartheta_i) - inv(\vartheta)) \cdot m; \ \delta_1 + \delta_2 = \frac{x_1 + x_2}{m} \end{cases}. \tag{5}$$

The designer has then to split the sum of profile shifting coefficients $\delta_1 + \delta_2$ into two terms, to be applied to the pinion ($\delta_1$) and the wheel ($\delta_2$), respectively. For this purpose, the designer may take advantage of Equation (6), based on the suggestions of Reference [26–28]: In particular, for a fast selection (starting point), Reference [26], highlights that the proposed Equation (6) allows, with a certain approximation, the same specific sliding at the tooth base for the pinion and the gear.

$$\begin{cases} \delta_1 \approx \frac{\delta_1 + \delta_2}{2} \cdot \left( \frac{z_1}{100} + \tau - \tau \cdot \frac{z_1}{100} \right) + 0.5 \cdot \left( 1 - \frac{z_1}{100} \right) \cdot (1 - \tau) \\ \delta_2 = (\delta_1 + \delta_2) - \delta_1 \end{cases}. \tag{6}$$

It is then possible to determine the modified tooth shape and, in particular:

- the coefficient $k$, which, in turn, makes it possible to evaluate the required reduction of addendum (tooth head cutting). To ensure the same clearance (see Figure 4) with the modified working geometry (Equation (7)):

$$k = \left| \frac{I_i - I - (x_1 + x_2)}{m} \right|; \tag{7}$$

- the working addendum $a_i$ and dedendum $d_i$ [mm] (Equation (8)):

$$\begin{cases} a_{i\_1,2} = a + x_{1,2} - k \cdot m \\ d_{i\_1,2} = d - x_{1,2} \end{cases}; \tag{8}$$

- the actual minimum number of teeth, considering profile shifting. to avoid under-cut (Equation (9)):

$$z_{i\_1\_min} = \frac{2 \cdot (a - x_1)}{m \cdot sin^2 \vartheta}; \tag{9}$$

- the tooth thickness si_1,2 [mm], evaluated at the reference diameter (Equation (10)):

$$s_{i\_1,2} = m \cdot \frac{\cos(\vartheta)}{\cos(\vartheta_i)} \cdot \left[ \frac{\pi}{2} + 2 \cdot \delta_{1,2} \cdot \tan(\vartheta) - z_{1,2} \cdot (inv(\vartheta_i) - inv(\vartheta)) \right]. \tag{10}$$

An example of tooth shape evolution, as a function of profile shifting coefficient is shown in Figure 5.

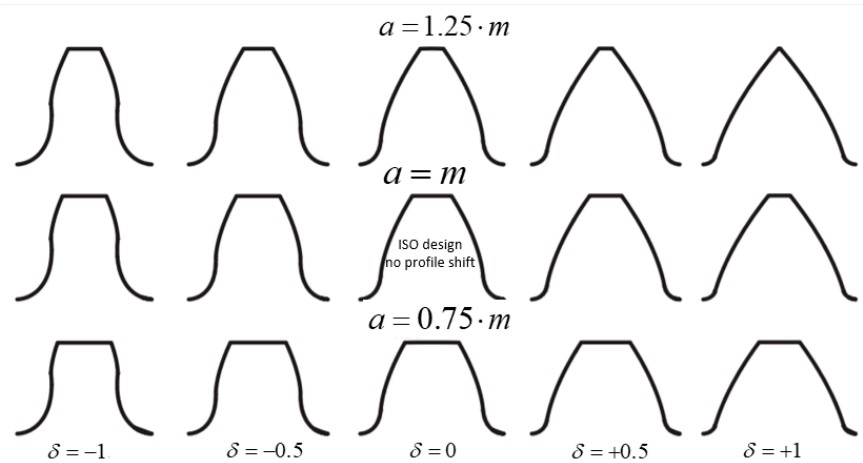

**Figure 5.** Effect of profile shifting on tooth shape.

Once the geometrical parameters have been defined, it is possible to proceed with gear calculation against tooth bending and contact stresses. This procedure leads to the final accurate estimation of the gear module. The maximum stress must be lower than the allowable one: the resistance conditions for bending and pitting are reported in Equations (11) and (12), respectively (see Reference [28]):

$$\sigma_{\text{max\_bending}} = \frac{\left(\frac{2000 \cdot M_t}{D}\right) \cdot q \cdot L}{b \cdot m \cdot \varepsilon \cdot \eta_d} \leq \sigma_{\text{lim\_bending}}, \tag{11}$$

$$\begin{cases} p_{\text{max\_pitting}} = f \cdot \sqrt{\dfrac{\frac{\left(\frac{2000 \cdot M_t}{D}\right) \cdot L}{b \cdot m} \cdot \left(\frac{1}{z_1} + \frac{1}{z_2}\right)}{\eta_d \cdot \eta_E}} \leq p_{\text{lim\_pitting}} = \dfrac{2.5 \cdot H_d}{\sqrt[6]{n \cdot h}} \\ f = \sqrt{\dfrac{0.417^2 \cdot 2}{\frac{1}{2} \cdot \left(\frac{1}{E_1} + \frac{1}{E_2}\right) \cdot sen(\vartheta) \cdot \cos(\vartheta)}} \end{cases} \tag{12}$$

With reference to these formulas, the following parameters need to be considered, according to Reference [28]:

- q is the tooth form factor (with load applied at the outer point of single tooth pair contact): it takes tooth shape into account (Table 2);
- L is the dynamic factor, which accounts for load increments due to dynamic effects upon load application (Table 3);
- $\varepsilon$ is the factor considering load sharing between more than one tooth pair (Table 4);
- $\eta_d$ is the factor that takes speed effect into account (Table 5);
- $\eta_E$ is the factor depending on lubrication condition (Table 6). For high speed gears (v > 10 m/s) oil viscosity can be selected between 46 cst and 100 cst (increasing with the applied load); for highly loaded gears (usually not at high speed), oil viscosity is within 150 cst and 680 cst (increasing with load); for v < 2 m/s, even permanent grease lubrication may be chosen. Wear resistance generally increases with viscosity; however, the higher viscosity, the higher power losses and temperature increase (for service temperature greater than 60 °C, it is suggested to cool the oil);
- the parameter $f$ [N$^{0.5}$/mm], for steel on steel gears (Young's modulus E = 200 GPa) and $\vartheta = 20°$, takes the value of 473 N$^{0.5}$/mm; when profile shifting is applied, f must be calculated with $\vartheta_i$ instead of $\vartheta$ (with positive profile shift, $\vartheta_i > \vartheta$; therefore, it is possible to achieve benefits in terms of maximum contact stress value reduction).

**Table 2.** Values of tooth factor q, according to Reference [28], for cutting pressure angle of 20° and modular (ISO 53) proportion; for stub proportion: $q_{stub} = q \cdot 0.8$.

| z Number of Teeth | δ Profile Shift Coefficient [-] | | | | | | | | | | |
|---|---|---|---|---|---|---|---|---|---|---|---|
| | −0.5 | −0.4 | −0.3 | −0.2 | −0.1 | 0 | 0.1 | 0.2 | 0.3 | 0.4 | 0.5 |
| 10 | * | * | * | * | * | * | 4.33 | 3.68 | 3.27 | 3.02 | 2.87 |
| 12 | * | * | * | * | * | 4.75 | 3.84 | 3.37 | 3.06 | 2.88 | 2.79 |
| 15 | * | * | * | * | 4.55 | 3.9 | 3.45 | 3.15 | 2.92 | 2.78 | 2.7 |
| 20 | * | * | 4.85 | 4.23 | 3.76 | 3.37 | 3.12 | 2.92 | 2.76 | 2.66 | 2.6 |
| 25 | 4.9 | 4.44 | 4.04 | 3.71 | 3.41 | 3.14 | 2.98 | 2.82 | 2.69 | 2.6 | 2.55 |
| 30 | 4.13 | 3.87 | 3.64 | 3.45 | 3.24 | 3.05 | 2.89 | 2.75 | 2.64 | 2.56 | 2.52 |
| 35 | 3.83 | 3.65 | 3.45 | 3.29 | 3.12 | 2.97 | 2.82 | 2.7 | 2.6 | 2.54 | 2.49 |
| 40 | 3.68 | 3.51 | 3.34 | 3.18 | 3.04 | 2.9 | 2.77 | 2.66 | 2.57 | 2.52 | 2.47 |
| 50 | 3.47 | 3.32 | 3.17 | 3.03 | 2.91 | 2.8 | 2.69 | 2.6 | 2.53 | 2.48 | 2.44 |
| 60 | 3.34 | 3.2 | 3.07 | 2.95 | 3.83 | 2.72 | 2.64 | 2.57 | 2.5 | 2.46 | 2.42 |
| 80 | 3.14 | 3.02 | 2.92 | 2.81 | 2.7 | 2.63 | 2.57 | 2.51 | 2.46 | 2.42 | 2.39 |
| 100 | 3.02 | 2.91 | 2.82 | 2.72 | 2.64 | 2.57 | 2.53 | 2.48 | 2.43 | 2.4 | 2.38 |
| 150 | 2.84 | 2.76 | 2.69 | 2.62 | 2.55 | 2.5 | 2.46 | 2.43 | 2.39 | 2.37 | 2.36 |

* undercut: not possible.

**Table 3.** Values of load application factor *L*, according to Reference [28].

| | |
|---|---|
| *L* = [1–1.25] | Regular Motion—No Shocks |
| *L* = [1.25–1.5] | Limited shocks |
| *L* = [1.5–1.75] | Small shocks |
| *L* = [1.75–2.5] | Big shocks |

**Table 4.** Values of load sharing coefficient ε, according to Reference [28].

| z | ε | z | ε |
|---|---|---|---|
| 10 | 1.25 | 25 | 1.33 |
| 11 | 1.26 | 30 | 1.36 |
| 12 | 1.26 | 35 | 1.39 |
| 13 | 1.27 | 40 | 1.42 |
| 14 | 1.27 | 45 | 1.44 |
| 15 | 1.28 | 50 | 1.47 |
| 16 | 1.28 | 60 | 1.52 |
| 17 | 1.29 | 70 | 1.58 |
| 18 | 1.29 | 80 | 1.63 |
| 19 | 1.30 | 90 | 1.69 |
| 20 | 1.31 | 100 | 1.75 |

**Table 5.** Values of speed effect coefficient $\eta_d$ according to Reference [28]: (a) gears with quality grade from 6 to 8 [$\eta_d = 5.6/(5.6 + v^{0.5})$]; (b) gears with quality grade from 4 to 7 (for quality grade definition, see the next section).

| (a) | | |
|---|---|---|
| **v [m/s]** | $\eta_d$ | $1/\eta_d$ |
| 5 | 0.71 | 1.40 |
| 6 | 0.70 | 1.44 |
| 7 | 0.68 | 1.47 |
| 8 | 0.66 | 1.51 |
| 9 | 0.65 | 1.54 |
| 10 | 0.64 | 1.56 |
| 11 | 0.63 | 1.59 |
| 12 | 0.62 | 1.62 |

| (b) | | | | | |
|---|---|---|---|---|---|
| **v [m/s]** | $\eta_d$ | $1/\eta_d$ | **v [m/s]** | $\eta_d$ | $1/\eta_d$ |
| 5 | 1 | 1 | 13 | 0.83 | 1.2 |
| 6 | 1 | 1 | 14 | 0.82 | 1.22 |
| 7 | 1 | 1 | 15 | 0.81 | 1.24 |
| 8 | 1 | 1 | 16 | 0.8 | 1.25 |
| 9 | 1 | 1 | 17 | 0.79 | 1.26 |
| 10 | 1 | 1 | 18 | 0.79 | 1.27 |
| 11 | 1 | 1 | 19 | 0.78 | 1.28 |
| 12 | 1 | 1 | 20 | 0.77 | 1.3 |

**Table 6.** Values of lubrication condition factor $\eta_E$, according to Reference [28].

| **ISO VG [cst]** | **Engler [°]** | $\eta_E$ |
|---|---|---|
| 32 | 4.5 | 0.7 |
| 46 | 6 | 0.8 |
| 68 | 8.9 | 0.9 |
| 100 | 13.2 | 1 |
| 150 | 19.8 | 1.1 |
| 220 | 29 | 1.2 |
| 320 | 42.2 | 1.3 |
| 460 | 60.7 | - |
| 680 | 89.8 | - |

As for the expected working hours *h*, reference values can be considered as:

- $h$ = 40,000 to 150,000 for applications running 24 h per day (like turbines or important production machines);
- $h$ = 20,000 to 30,000 for applications running 8 h per day (standard production machines);
- $h$ = 5000 to 15,000 for applications running some hours per day (automotive or lifting devices);
- $h$ = 500 to 1500 for limited running;

Finally, allowable stresses may be worked out from the values reported in Table 7: Bending strength also takes stress concentration at tooth root into account for a standard 0.25·m fillet radius (Figure 4).

**Table 7.** Values of allowable stresses, according to Reference [28], in the case of pulsating load (for alternate load: $\sigma_{\text{lim\_bending}}$ ·0.7).

| Material | Ultimate Stress [MPa] | $\sigma_{\text{lim\_bending}}$ [MPa] | $H_d$ [MPa] |
|---|---|---|---|
| Cast iron | 180 to 260 | 40 to 55 | 1700 to 2100 |
| Structural steel | 500 to 600 | 90 to 100 | 1500 to 1800 |
| Carbon steel | 500 to 700 | 110 to 140 | 1600 to 2100 |
| Quenched and tempered steel | 700 to 1000 | 135 to 200 | 1850 to 2600 <br><br> • Surface hardened ~52 HRC: 5200; <br><br> • Nitrided > 700 HV: 5500 |
| Case hardening steel | 600 to 1200 | 125 to 200 | 2500 <br><br> • Surface hardened > 56 HRC: 6500 |
| Bronze | 200 to 320 | 80 to 120 | 900 to 1200 |
| Plastic material | 150 | 35 | 350 |

In order to correctly determine the most suitable module value (after its initial rough estimation, based on the raw formula of Equation (2)), the inequalities in Equations (11) and (12) have been inverted, according to Equations (13) and (14), so that module thresholds are yielded as follows:

$$m_{bending} \geq \sqrt[3]{\frac{2000 \cdot M_{t\_1,2} \cdot q_{1,2} \cdot L}{z_{1,2} \cdot \lambda \cdot \varepsilon_{1,2} \cdot \eta_{d\_1,2} \cdot \sigma_{\text{lim\_bending\_1,2}}}}, \tag{13}$$

$$m_{pitting} \geq 0.69 \cdot \sqrt[3]{\frac{f_{1,2}^2 \cdot L \cdot M_{t\_1,2} \cdot \left(\frac{1}{z_1} + \frac{1}{z_2}\right)}{\eta_{d\_1,2} \cdot \eta_E \cdot H_{d\_1,2}^2 \cdot \lambda \cdot z_{1,2}}} \cdot \sqrt[9]{n_{1,2} \cdot h}. \tag{14}$$

All the proposed steps are particularly suitable for automatic computation by electronic datasheets and may also be used for optimization purposes, as in the numerical example in Section 5. The described procedure is also available for more sophisticated optimization strategies, for instance by Genetic Algorithms, as proposed in Reference [29]. For the sake of clarity and readability, a flowchart collecting all the steps is provided in the Appendix A.

In addition, the proposed Equations can be applied not only to cylindrical spur gear teeth but also to helical teeth, according to Reference [28]. Let β be the helix angle (β > 0 for right-handed helix); it is then possible to evaluate the parameters on two different planes: *normal plane*, subscript $_n$, which is the plane being perpendicular with respect to the tooth axis (axis of the cutting tool); and *circumferential plane*, subscript $_c$, which is perpendicular to the gear axis. The following formulas apply [28], being $m_n$ the normal module, according to ISO54, $\vartheta_n$ the normal pressure angle (usually 20°) and $z$ the number of teeth:

$m_c = \frac{m_n}{cos\beta}$                          circumferential module [mm]

$\tan(\vartheta_c) = \frac{\tan(\vartheta_n)}{cos\beta}$                for the evaluation of the circumferential pressure angle [°]

$z^* = \frac{z}{[cos(\beta)]^3}$                  virtual number of teeth of an equivalent spur gear in the normal plane

$D = z \cdot m_c$                               pitch diameter [mm]

$\varepsilon_a^* = \frac{g^*}{t_n} = \frac{g^*}{\pi \cdot m_n}$               virtual transverse contact ratio, calculated with $z$ * and $m_n$

For tooth design, it is possible to apply both ISO 53 or Stub proportion, based on $m_n$. Correction selection may be addressed by the same formula for spur gears, introducing $z^*$ and $\vartheta_c$ instead of $z$ and $\vartheta$. For design purposes, to ensure sufficient strength against bending, $q$ is a function of $z^*$ instead of $z$, whereas $m_n$ is used instead of $m$, and $\varepsilon$ can be considered equal to 2. Finally, when addressing design against pitting, *1.25·b* may replace $b$ with regard to face width.

## 3. Gear Lubrication

Reducing friction, increasing efficiency, reducing wear and contact fatigue of the interacting tooth surfaces, and improving durability can be regarded as the main purposes of gear lubrication. Lubrication also facilitates heat exchanges between equipment, dissipating the heat produced by friction, in order to mitigate the power losses. Lubricants are essential to transmissions and can therefore be considered as a real mechanical unit. As a function of the running speed, it is possible to categorize friction within three levels. (i) The first one is boundary friction, where the tooth flanks are only separated by a boundary layer consisting of chemical reaction products. This layer, being a few nanometers thick, is intended to prevent metal to metal contact. (ii) The second one is mixed friction, involving the tooth flanks being partially separated by a lubricant film. Both liquid and dry friction occur at the same time: where the surfaces are in contact, friction turns to be of the boundary type. (iii) Finally, fluid friction (hydrodynamic) occurs when tooth flanks are completely separated by a lubricant film, which involves elasto-hydrodynamic (EHD) lubrication. In the presence of precise gears, with fast running speed (v > 20 m/s) the regime (iii) can be reached: the thickness of the oil film $h_c$ (see Reference [27,30–32]) is comparable to the roughness of the teeth $(R_{a1} + R_{a2})/2$ and, in the scenario of $h_c > (R_{a1} + R_{a2})/2$, the EHD lubrication occurs. Conversely, for standard solutions, i.e., industrial gearboxes, regime (i) and (ii) is reached; therefore, additives must be properly used in order to improve the friction and wear behavior in the presence of dry (or mixed) friction.

Several studies dealing with gear lubrication have been performed, addressing in recent years EHD lubrication and analyses or comparisons between mineral and synthetic customized lubricants. Interesting studies about the effects of lubrication on gear performance are reported in Reference [33,34]; the effects of different types of lubricants, as well as a comparative overview of several gear oils in mixed and boundary film lubrication, are reported in Reference [35–37]. According to Reference [38], several tests have been performed at the Gear Research Center (FZG) of TU Munich to develop gear transmission fluids, based on water and plant extract, thus avoiding using non-biodegradable fossil raw material. Similar interesting studies are reported in References [39,40]. Recently, the use of the finite volume Computational Fluid Dynamics modeling (CFD) method has been adopted to study and optimize the lubrication process [41–43]. However, numerical models dealing with EHD contacts are often computationally expensive, which makes them unsuitable to assist gear design [44–46]. A further refinement may be achieved by these tools, but this is beyond the scopes of the present review study. Gear transmission designer has to select the most proper lubricant in the market and, for this purpose, has to follow some practical design rules based on the analyses of the power losses [47–51], such as:

- Selecting reference working temperature within 60 to 110 °C;
- Selecting modern lubricants, which have a base and appropriate additives;
- Oils are commonly classified, based on their V.I. (viscosity index, ISO 2909), which indicates the behavior of viscosity versus temperature: standard quality oils has V.I. = 95 to 105, whereas high quality oils has V.I. > 150;
- If T < 20 °C or T > 110 °C, then synthetic oils have to be used;
- Most common additives are EP (extreme pressure), AW (antiwear), V.I. (improver);
- A compromise is needed between: (i) gear resistance to scuffing and pitting increase with viscosity (high viscosity desired); (ii) damping capacity increase with viscosity (high viscosity desired); and (iii) friction losses increase with viscosity, so that temperature also increases with viscosity (low viscosity desired).



A comparison involving some viscosity classifications (basis V.I. = 100) is shown in Figure 6a, together with the so-called Ubellohde diagram (log-log scale, Figure 6b), where viscosity versus temperature trends are displayed for different gearbox oils (a: mineral; b: Poly-α-oleofin-based; c: Polyglycol-based).

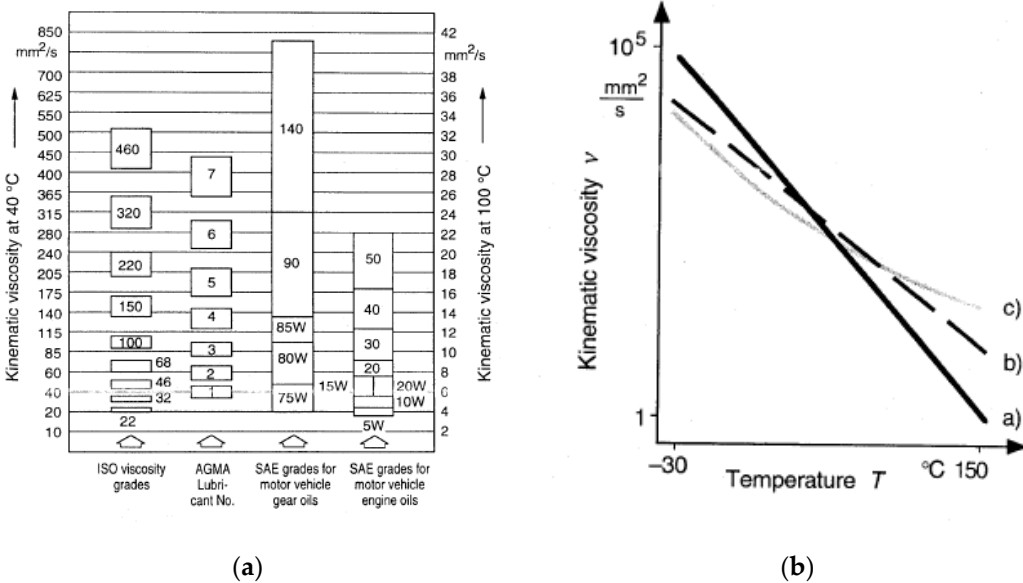

(**a**)                                                                                    (**b**)

**Figure 6.** Mineral oil selection. Example: gear with reference diameter $D$ = 200 mm, $n$ = 500 rpm, working temperature of 90 °C; from (**a**): viscosity required at working temperature equal to ISO VG22; from (**b**): viscosity required at 40 °C equal to ISO VG 150.

Regarding synthetic products, it is generally possible to properly select the lubricant, according to technical data provided in the datasheet in the market. Two examples of lubrication labels applied in gearboxes are shown in the following pictures of Figure 7: lubrication via synthetic oil (ISO VG 220) for gears and lubrication via synthetic long-term gear grease.

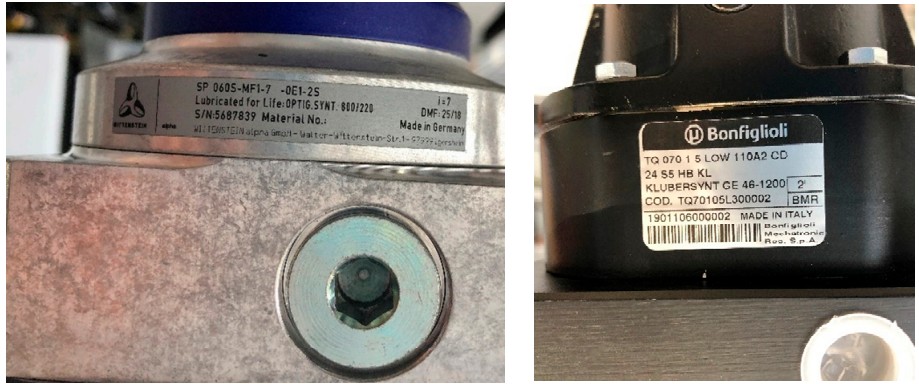

**Figure 7.** Example of labels dedicated to lubrication in gearboxes.

## 4. Gear Quality and Tolerances

Following proper module calculation and lubricant selection, as described in Sections 2 and 3, gear designer has to select the right quality and tolerances, according to the precision grade of the gear. This operation can be done by leveraging International Standards, such as ISO1328 [52] (formerly UNI7880 [53]) or DIN3967 [54]. With regard to precision grade selection, reference can be made to Table 8, where quality class is provided, depending on the industrial application and on the linear speed v [m/s] of the gear [55].

**Table 8.** ISO or DIN quality class.

|  | 1 | 2 | 3 | 4 | 5 | 6 | 7 | 8 | 9 | 10 | 11 | 12 |
|---|---|---|---|---|---|---|---|---|---|---|---|---|
| Master gears | ▓ | ▓ |  |  |  |  |  |  |  |  |  |  |
| Precision gears—mechanical dividers |  | ▓ | ▓ | ▓ |  |  |  |  |  |  |  |  |
| High speed gears |  |  | ▓ | ▓ | ▓ |  |  |  |  |  |  |  |
| Gear for precise gearboxes |  |  |  | ▓ | ▓ | ▓ |  |  |  |  |  |  |
| Gear for gearboxes |  |  |  |  | ▓ | ▓ |  |  |  |  |  |  |
| Gear for generic mechanics |  |  |  |  | ▓ | ▓ | ▓ | ▓ |  |  |  |  |
| Low speed gears |  |  |  |  |  |  |  | ▓ | ▓ | ▓ | ▓ | ▓ |
|  | 1 | 2 | 3 | 4 | 5 | 6 | 7 | 8 | 9 | 10 | 11 | 12 |

| v [m/s] | | |
|---|---|---|
| >20 | (grades 3–5) |
| 8 . . . 20 | (grades 5–7) |
| 3 . . . 8 | (grades 7–9) |
| 0 . . . 3 | (grades 9–11) |

Once the quality class has been selected, the geometrical and dimensional tolerances can be set, based on the data reported in Table 9, which reports standard tolerance intervals in agreement with Standards [52,53].

**Table 9.** Dimensional and geometrical tolerances, according to Reference [52,53].

| Tolerances on the Hub Hole Diameter, Shaft Diameter and Wheel External Diameter | | | | | | | | | | | | |
|---|---|---|---|---|---|---|---|---|---|---|---|---|
| **Precision Grade** | **1** | **2** | **3** | **4** | **5** | **6** | **7** | **8** | **9** | **10** | **11** | **12** |
| HUB — Diameter Tolerance | IT 4 | IT 4 | IT 4 | IT 4 | IT 5 | IT 6 | IT 7 | IT 7 | IT 8 | IT 8 | IT 8 | IT 8 |
| HUB — Form Tolerance | IT 1 | IT 2 | IT 3 | | | | | | | | | |
| SHAFT — Diameter Tolerance | IT 4 | IT 4 | IT 4 | IT 4 | IT 5 | IT 5 | IT 6 | IT 6 | IT 7 | IT 7 | IT 8 | IT 8 |
| SHAFT — Form Tolerance | IT 1 | IT 2 | IT 3 | | | | | | | | | |
| Tolerance on External Diameter | IT 6 | IT 6 | IT 7 | IT 7 | IT 7 | IT 8 | IT 8 | IT 8 | IT 9 | IT 9 | IT 11 | IT 11 |

| Reference Surface Radial and Axial Runout Tolerance | | | | | |
|---|---|---|---|---|---|
| **Reference Diameter** | **Precision Grade** | | | | |
| **mm** | **1 and 2** | **3 and 4** | **5 and 6** | **7 and 8** | **from 9 to 12** |
| OVER — TO | Radial and axial runout in μm | | | | |
| - — 125 | 2.8 | 7 | 11 | 18 | 28 |
| 125 — 400 | 3.6 | 9 | 14 | 22 | 36 |
| 400 — 800 | 5.0 | 12 | 20 | 32 | 50 |
| 800 — 1600 | 7.0 | 18 | 28 | 45 | 71 |
| 1600 — 2500 | 10.0 | 25 | 40 | 63 | 100 |
| 2500 — 4000 | 16.0 | 40 | 63 | 100 | 160 |

An example of pinion drawing, to be used in a rotary transfer machine, where it has the specific role of moving the main rotary table [56], is reported in Figure 8. The main features related to this industrial application are: $M_t$ = 2.75 kNm, $\tau$ = 0.2, $n$ = 75 rpm, theoretical wheelbase 340 mm, material: 18NiCrMo5 (hardening 58–60 HRC).

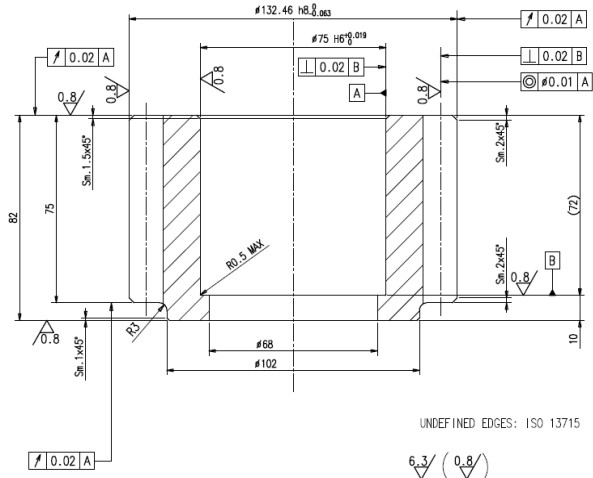

| CYLINDRICAL SPUR GEAR – STUB DESIGN – addendum=0.8 × m | |
|---|---|
| N° OF TEETH | 14 |
| GEAR MODULE | 8 |
| PRESSURE ANGLE | 20° |
| RACK TYPE CUTTER | UNI 6587 |
| PROFILE SHIFT COEFFICIENT | 0.5 |
| REFERENCE CRICLE (pitch circle) | 112 mm |
| ADDENDUM CIRCLE (head diameter) | 132.46 mm |
| DEDENDUM CIRCLE | 104 mm |
| BASE CIRCLE | 105.25 mm |
| SPAN MEASURE Wk (over k=2 teeth) | 39.73 mm $_{-0.16}^{-0.08}$ |
| QUALITY CLASS (UNI 7880) | 7 FH |
| N° OF TEETH OF THE GEAR | 71 |
| WORKING WHEELBASE | 344 mm |
| TEETH ROUGHNESS | 0.8 |
| NORMAL BACKLASH Jn (min–MAX) referred to the nominal working wheelbase | 0.18 mm – 0.36 mm |
| DEPTH OF HARDENING | 1.6 – 1.8 mm |

**Figure 8.** Example of pinion drawing.

The last important parameters to be calculated and reported in the gear table are: (i) the span measurement $W_k$ (mm) and its tolerance, and (ii) the normal backlash $j_n$ (mm) between the mating gears (referred to the nominal working wheelbase). The theoretical span measurement can be calculated by Equation (15), where the $k$ pedix indicates the number of teeth between the measuring gauges. Its tolerance is a function of single pitch deviation $f_{pt}$ (μm) reported in Table 10, according to letters from C to S [52,53] (practical values between, *E* and *L*: FG, GH, or HJ suitable for generic mechanics).

$$\begin{cases} W_k = m \cdot \cos(\vartheta) \cdot [(k-0.5) \cdot \pi + z \cdot inv(\theta) + 2 \cdot \delta \cdot \tan(\vartheta)] \\ k = 0.5 + z \cdot \frac{\vartheta}{180} - \frac{2 \cdot \delta \cdot \tan(\vartheta)}{180} \end{cases} . \tag{15}$$

**Table 10.** Single pitch deviation, according to Reference [52,53].

| Single Pitch Deviation | | | | | | | | | | | | | | |
|---|---|---|---|---|---|---|---|---|---|---|---|---|---|---|
| Reference Diameter | | Module | | Precision Grade | | | | | | | | | | |
| mm | | mm | | 1 | 2 | 3 | 4 | 5 | 6 | 7 | 8 | 9 | 10 | 11 | 12 |
| over | to | from | to | | | | | | $f_{pt}$ μm | | | | | | |
| - | 125 | 1 | 3.5 | 1 | 1.6 | 2.5 | 4 | 6 | 10 | 14 | 20 | 28 | 40 | 56 | 80 |
| | | 3.5 | 6.3 | 1.2 | 2 | 3.2 | 5 | 8 | 13 | 18 | 25 | 36 | 50 | 71 | 100 |
| | | 6.3 | 10 | 1.4 | 2.2 | 3.6 | 5.5 | 9 | 14 | 20 | 28 | 40 | 56 | 80 | 112 |
| 125 | 400 | 1 | 3.5 | 1.1 | 1.8 | 2.8 | 4.5 | 7 | 11 | 16 | 22 | 32 | 45 | 63 | 90 |
| | | 3.5 | 6.3 | 1.4 | 2.2 | 3.6 | 5.5 | 9 | 14 | 20 | 28 | 40 | 56 | 80 | 112 |
| | | 6.3 | 10 | 1.6 | 2.5 | 4 | 6 | 10 | 16 | 22 | 32 | 45 | 63 | 90 | 125 |
| | | 10 | 16 | 1.8 | 2.8 | 4.5 | 7 | 11 | 18 | 25 | 36 | 50 | 71 | 100 | 140 |
| | | 16 | 25 | 2.2 | 3.6 | 5.5 | 9 | 14 | 22 | 32 | 45 | 63 | 90 | 125 | 180 |
| 400 | 800 | 1 | 3.5 | 1.2 | 2 | 3.2 | 5 | 8 | 13 | 18 | 25 | 36 | 50 | 71 | 100 |
| | | 3.5 | 6.3 | 1.4 | 2.2 | 3.6 | 5.5 | 9 | 14 | 20 | 28 | 40 | 56 | 80 | 112 |
| | | 6.3 | 10 | 1.8 | 2.8 | 4.5 | 7 | 11 | 18 | 25 | 36 | 50 | 71 | 100 | 140 |
| | | 10 | 16 | 2 | 3.2 | 5 | 8 | 13 | 20 | 28 | 40 | 56 | 80 | 112 | 160 |
| | | 16 | 25 | 2.5 | 4 | 6 | 10 | 16 | 25 | 36 | 50 | 71 | 100 | 140 | 200 |
| | | 25 | 40 | 3.2 | 5 | 8 | 13 | 20 | 32 | 45 | 63 | 90 | 125 | 180 | 250 |
| 800 | 1600 | 1 | 3.5 | 1.2 | 2 | 3.6 | 5.5 | 9 | 14 | 20 | 28 | 40 | 56 | 80 | 112 |
| | | 3.5 | 6.3 | 1.6 | 2.5 | 4 | 6 | 10 | 16 | 22 | 32 | 45 | 63 | 90 | 125 |
| | | 6.3 | 10 | 1.8 | 2.8 | 4.5 | 7 | 11 | 18 | 25 | 36 | 50 | 71 | 100 | 140 |
| | | 10 | 16 | 2 | 3.2 | 5 | 8 | 13 | 20 | 28 | 40 | 56 | 80 | 112 | 160 |
| | | 16 | 25 | 2.5 | 4 | 6 | 10 | 16 | 25 | 36 | 50 | 71 | 100 | 140 | 200 |
| | | 25 | 40 | 3.2 | 5 | 8 | 13 | 20 | 32 | 45 | 63 | 90 | 125 | 180 | 250 |

**Table 10.** *Cont.*

| Single Pitch Deviation | | | | | | | | | | | | | | | |
|---|---|---|---|---|---|---|---|---|---|---|---|---|---|---|---|
| Reference Diameter | | Module | | Precision Grade | | | | | | | | | | | |
| mm | | mm | | 1 | 2 | 3 | 4 | 5 | 6 | 7 | 8 | 9 | 10 | 11 | 12 |
| over | to | from | to | $f_{pt}$ μm | | | | | | | | | | | |
| 1600 | 2500 | 1 | 3.5 | 1.6 | 2.5 | 4 | 6 | 10 | 16 | 22 | 32 | 45 | 63 | 90 | 125 |
| | | 3.5 | 6.3 | 1.8 | 2.8 | 4.5 | 7 | 11 | 18 | 25 | 36 | 50 | 71 | 100 | 140 |
| | | 6.3 | 10 | 2 | 3.2 | 5 | 8 | 13 | 20 | 28 | 40 | 56 | 80 | 112 | 160 |
| | | 10 | 16 | 2.2 | 3.6 | 5.5 | 9 | 14 | 22 | 32 | 45 | 63 | 90 | 125 | 180 |
| | | 16 | 25 | 2.8 | 4.5 | 7 | 11 | 18 | 28 | 40 | 56 | 80 | 112 | 160 | 224 |
| | | 25 | 40 | 3.6 | 5.5 | 9 | 14 | 22 | 36 | 50 | 71 | 100 | 140 | 200 | 280 |
| 2500 | 4000 | 1 | 3.5 | 1.8 | 2.8 | 4.5 | 7 | 11 | 18 | 25 | 36 | 50 | 71 | 100 | 140 |
| | | 3.5 | 6.3 | 2 | 3.2 | 5 | 8 | 13 | 20 | 28 | 40 | 56 | 80 | 112 | 160 |
| | | 6.3 | 10 | 2.2 | 3.6 | 5.5 | 9 | 14 | 22 | 32 | 45 | 63 | 90 | 125 | 180 |
| | | 10 | 16 | 2.5 | 4 | 6 | 10 | 16 | 25 | 36 | 50 | 71 | 100 | 140 | 200 |
| | | 16 | 25 | 2.8 | 4.5 | 7 | 11 | 18 | 28 | 40 | 56 | 80 | 112 | 160 | 224 |
| | | 25 | 40 | 3.6 | 5.5 | 9 | 14 | 22 | 36 | 50 | 71 | 100 | 140 | 200 | 280 |

| | | | |
|---|---|---|---|
| $C = +1\,f_{pt}$ | $G = -6\,f_{pt}$ | $L = -16\,f_{pt}$ | $R = -40\,f_{pt}$ |
| $D = 0$ | $H = -8\,f_{pt}$ | $M = -20\,f_{pt}$ | $S = -50\,f_{pt}$ |
| $E = -2\,f_{pt}$ | $J = -10\,f_{pt}$ | $N = -25\,f_{pt}$ | |
| $F = -4\,f_{pt}$ | $K = -12\,f_{pt}$ | $P = -32\,f_{pt}$ | |

With reference to the pinion in Figure 8, $k$ corresponds to 2 teeth. Moreover, based on the quality class 7 (FH), the tolerance of the span measurement ($W_k = 39.73$ mm) ranges from $E_{ws,1} = -4\,f_{pt}$ to $E_{wi,1} = -8\,f_{pt}$ (which yields the range (80 μm to 160 μm)). Finally, estimating the same values for the wheel makes it possible to estimate the normal backlash $j_n$ (mm), according to Equation (16):

$$\begin{cases} j_{n\_\min} = -(E_{ws,1} + E_{ws,2}) \\ j_{n\_\max} = -(E_{wi,1} + E_{wi,2}) \end{cases}. \tag{16}$$

The value of the backlash is a very important parameter since it affects both the stiffness and the lubrication condition of the mating gears [57,58].

A faster way to get a rough estimation value of the backlash is provided by the following formula: $j_n \sim 0.05 + 0.025 \cdot m + 0.01 \cdot v$. Another option to define the tolerances is given by DIN 3967 Standard [54]: The recommended procedure can be synthetized as follows. For generic mechanical applications, it is advisable to select the group number 8, in the series within c25 and e24 (most used: cd25), according to Figure 9 and Table 11. The formula for backlash range determination (referred to the nominal working wheelbase) is available in the following Equation (17):

$$\begin{cases} A_{sni1} - A_{sne1} = T_{sn1};\, A_{sni2} - A_{sne2} = T_{sn2} \\ j_{n\_\min} = -(A_{sne,1} + A_{sne,2}) \\ j_{n\_\max} = -(A_{sni,1} + A_{sni,2}) \end{cases}. \tag{17}$$

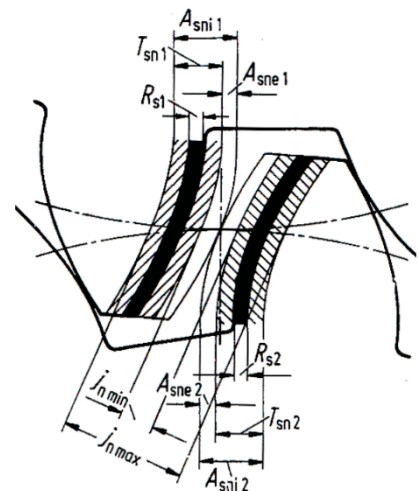

**Figure 9.** Definition of tooth thickness tolerances according to DIN3967 [54].

**Table 11.** Tooth thickness tolerances according to DIN3967 [54].

| Upper Tooth Thickness Allowance $A_{sne}$ in μm | | | | | | | | | | | |
|---|---|---|---|---|---|---|---|---|---|---|---|
| **Reference Diameter (mm)** | | **Allowance Series** | | | | | | | | | |
| over | up to | a | ab | b | bc | c | cd | d | e | f | g |
| - | 10 | −100 | −85 | −70 | −58 | −48 | −40 | −33 | −22 | −10 | −5 |
| 10 | 50 | −135 | −110 | −95 | −75 | −65 | −54 | −44 | −30 | −14 | −7 |
| 50 | 125 | −180 | −150 | −125 | −105 | −85 | −70 | −60 | −40 | −19 | −9 |
| 125 | 280 | −250 | −200 | −170 | −140 | −115 | −95 | −80 | −56 | −26 | −12 |
| 280 | 560 | −330 | −280 | −230 | −190 | −155 | −130 | −110 | −75 | −35 | −17 |
| 560 | 1000 | −450 | −370 | −310 | −260 | −210 | −175 | −145 | −100 | −48 | −22 |
| 1000 | 1600 | −600 | −500 | −420 | −340 | −290 | −240 | −200 | −135 | −64 | −30 |
| 1600 | 2500 | −820 | −680 | −560 | −460 | −390 | −320 | −270 | −180 | −85 | −41 |
| 2500 | 4000 | −1100 | −920 | −760 | −620 | −520 | −430 | −360 | −250 | −115 | −56 |
| Tooth Thickness Tolerance $T_{sn}$ in μm | | | | | | | | | | | |
| **Reference Diameter (mm)** | | **Allowance Series** | | | | | | | | | |
| over | up to | 21 | 22 | 23 | 24 | 25 | 26 | 27 | 28 | 29 | 30 |
| - | 10 | 3 | 5 | 8 | 12 | 20 | 30 | 50 | 80 | 130 | 200 |
| 10 | 50 | 5 | 8 | 12 | 20 | 30 | 50 | 80 | 130 | 200 | 300 |
| 50 | 125 | 6 | 10 | 16 | 25 | 40 | 60 | 100 | 160 | 250 | 400 |
| 125 | 280 | 8 | 12 | 20 | 30 | 50 | 80 | 130 | 200 | 300 | 500 |
| 280 | 560 | 10 | 16 | 25 | 40 | 60 | 100 | 160 | 250 | 400 | 600 |
| 560 | 1000 | 12 | 20 | 30 | 50 | 80 | 130 | 200 | 300 | 500 | 800 |
| 1000 | 1600 | 16 | 25 | 40 | 60 | 100 | 160 | 250 | 400 | 600 | 1000 |
| 1600 | 2500 | 20 | 30 | 50 | 80 | 130 | 200 | 300 | 500 | 800 | 1300 |
| 2500 | 4000 | 25 | 40 | 60 | 100 | 160 | 250 | 400 | 600 | 1000 | 1600 |

## 5. Numerical Example

In recent years, following the development of the E-mobility, the need for high power density has been more and more increased. Therefore, the goal of maximizing the transmittable power while simultaneously minimizing weight is a highly challenging task. Taking advantage of the

design methodology proposed in the previous Section, an optimal solution may be easily worked out. In particular, starting from some design parameters and specifications, the optimal gear geometry can be determined. Let us consider a case study with $I_i$ = 99 mm, $\tau$ = 0.5, $\lambda$ = 10, $n_1$ = 1500 rpm, $h$ = 20,000 h, for a gear made of case-hardened steel. The aforementioned procedure leads to the results in Table 12 (only design against pitting is considered). The same results are also reported in Figure 10 in terms of torque-to-mass ratio: it can be highlighted that such ratio exhibits a maximum for m = 4, which is therefore the most suitable value for gear module.

**Table 12.** Numerical example of optimization.

| $I$ [mm] | 99 | 97.5 | 96 | 99 | 99 |
|---|---|---|---|---|---|
| $\varepsilon_a$ [mm] | 1.49 | 1.46 | 1.44 | 1.66 | 1.74 |
| $m$ [mm] | 6 | 5 | 4 | 3 | 2 |
| $Z_1$ | 11 | 13 | 16 | 22 | 33 |
| $D_1$ [mm] | 66 | 65 | 64 | 66 | 66 |
| $\eta_d$ | 1.0 | 1.0 | 1.0 | 1.0 | 1.0 |
| $\eta_E$ | 1.0 | 1.0 | 1.0 | 1.0 | 1.0 |
| $\delta$ | 0.2 | 0.3 | 0.5 | 0.2 | 0.2 |
| $f$ [N$^{0.5}$ mm] | 473 | 453 | 438 | 473 | 473 |
| $H_d$ [MPa] | 6500 | 6500 | 6500 | 6500 | 6500 |
| Volume [mm$^3$] | 2,052,712 | 165,915 | 128,679 | 102,635 | 68,424 |
| Mass [kg] | 1.61 | 1.30 | 1.01 | 0.81 | 0.54 |
| Torque [Nm] | 322 | 284 | 236 | 161 | 107 |
| Torque/Mass [Nm/kg] | 199.8 | 218.1 | 233.6 | 199.8 | 199.2 |

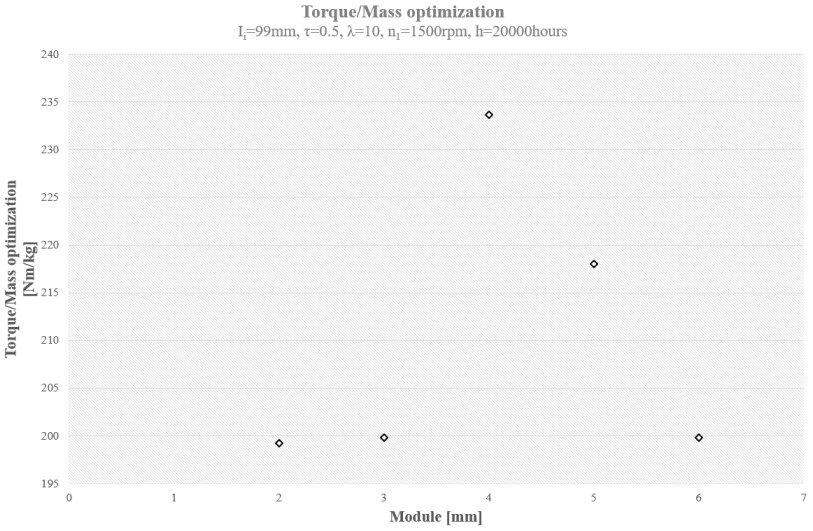

**Figure 10.** Torque/mass trend as a function of module *m*.

## 6. Conclusions

A review about the practical design of gears, their tolerances, and drawings is presented in the paper in order to propose a guideline for the designer. All the proposed formula, as well as the full methodology, has been used and in-field tested for several years in the framework of gear design in the machine tool industry. The described tools may be regarded as a comprehensive method to guide the engineer from the white paper to final design and gear drawing, following the required steps for gear parameter calculation and verification. From this point of view, this paper is particularly novel, as it is

a full-comprehensive collection of all the tools supporting gear design. Furthermore, the proposed method is particularly suitable for automatic computation by electronic datasheets.

**Funding:** This research received no external funding.

**Conflicts of Interest:** The authors declare no conflict of interest.

### Nomenclature (Alphabetic Order)

| | |
|---|---|
| $a$ | addendum: $(D_e - D)/2$ [mm] |
| $A_{sne}$ | minimum tooth error following the DIN3987 Standard |
| $A_{sni}$ | maximum tooth error following the DIN3987 Standard |
| b | $= \lambda \cdot m$: ($\lambda$ = [5–50], practical range $\lambda$ = [8–20]) face width [mm] |
| $d$ | dedendum: $(D - D_i)/2$ [mm] |
| $D$ | pitch (reference) circle (diameter): $_1$ = input gear—pinion; $_2$ = output gear—wheel [mm] |
| $D_b$ | $= D \cdot \cos(\vartheta)$: base circle [mm] |
| $D_e$ | addendum circle [mm] |
| $D_i$ | dedendum circle [mm] |
| $D_m$ | mean bearing diameter [mm] |
| $E$ | Young's modulus [MPa] |
| $E_{wi}$ | maximum tooth error |
| $E_{ws}$ | minimum tooth error |
| $f_{pt}$ | single pitch deviation [µm] |
| $g$ | arc of action (approach + recess) on the pitch circle [mm] $$= \tfrac{1}{2} \cdot m \cdot z_1 \left[ \sqrt{\left(\frac{1+2 \cdot a_1/m \cdot z_1}{\cos(\vartheta)}\right)^2 - 1} + \frac{1}{\tau} \cdot \sqrt{\left(\frac{1+2 \cdot a_2/m \cdot z_2}{\cos(\vartheta)}\right)^2 - 1} - \left(1 + \frac{1}{\tau}\right) \cdot \tan(\vartheta) \right]$$ In the case of working wheelbase different from theoretical wheelbase, $\vartheta_i$ must be considered. |
| $h$ | = a + d: tooth height [mm] |
| $h$ | daily hours of functioning [h] |
| $H_d$ | pressure limit [MPa] |
| $I$ | $= (D_1 + D_2)/2 = D_1(1/\tau + 1)/2 = D_2(1 + \tau)/2$: theoretical wheelbase (reference centre distance) [mm] |
| $I_i$ | working wheelbase (modified center distance) [mm] |
| $j_n$ | normal backlash [mm] |
| k | reduction of addendum due to profile shift |
| L | duty load coefficient |
| m | = D/z: module [mm] |
| $M_t$ | Torque moment [Nm] |
| n | revolutions per minute [rpm] |
| n | rotational speed ($_1$ = input; $_2$ = output) [rpm] |
| Q | amount of oil [mm$^3$/h] |
| q | shape coefficient (similar to Lewis one) |
| t | $= \pi \cdot m$: (circular) pitch [mm] |
| v | $= \omega \cdot (D/2000)$: linear speed [m/s] |
| $W_k$ | span measure [mm] |
| x | profile shifting [mm] |
| Y | Lewis coefficient |
| z | number of teeth ($_1$ = input; $_2$ = output) |
| $\varepsilon_\alpha$ | = g/t: (transverse) contact ratio [mm] |
| $\vartheta$ | cutting pressure angle (practical value: 20°) [°] |
| $\vartheta_i$ | working pressure angle [°] |
| $\tau$ | $= z_1/z_2$: transmission ratio |
| $\omega$ | $= 2\pi\, n/60$: rotational speed ($_1$ = input; $_2$ = output) [rad/s] |
| $\delta$ | profile shifting coefficient |
| $\eta_d$ | speed coefficient |
| $\eta_E$ | lubrication coefficient |
| $\sigma_{amm}$ | Admissible strength [MPa] |

## Appendix A

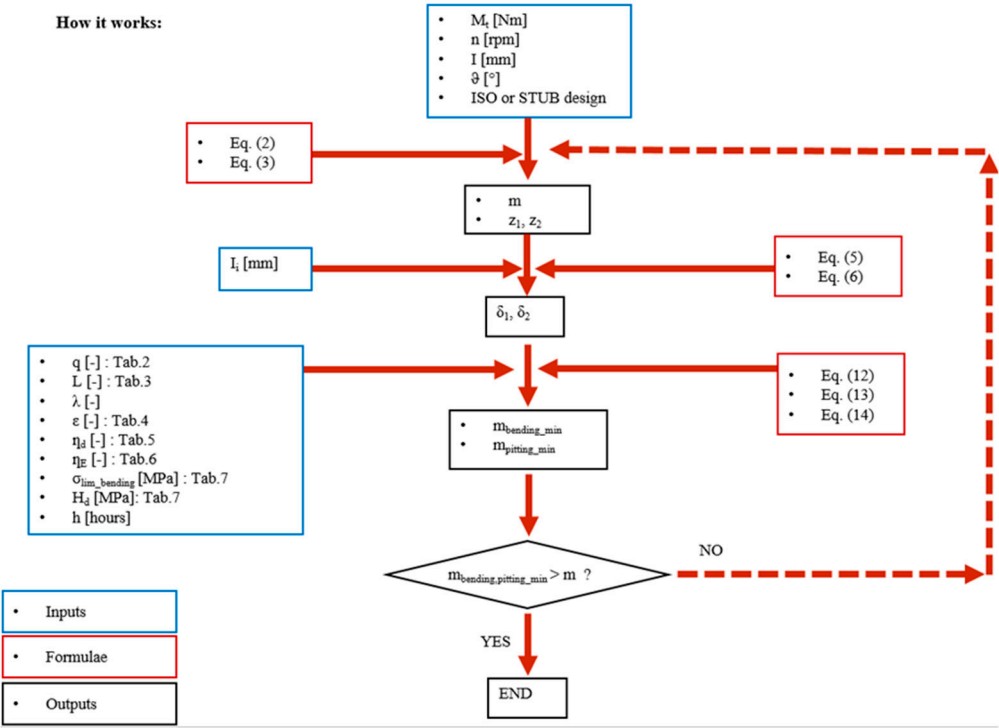

**Figure A1.** Flowchart collecting all the steps for cylindrical gear design.

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
