# Peer review of "A Practical Approach to Gear Design and Lubrication: A Review"

_lubricants, doi:10.3390/lubricants8090084_

Round 1
Reviewer 1 Report
Review (2020.06.16.) for the paper of Reference Number: lubricants-846-129
A Review About A Practical Approach to Gear Design and Lubrication
Dario Croccolo, Massimiliano De Agostinis, Giorgio Olmi, NicoloVincenzi
My comments and suggestions are:
- The title to be changed to: “A Review About A Practical Approach to Spur Gear Design and Lubrication”. The whole paper is only on the design of spur gears.
- V. Simon have published a big number of papers on the design, manufacture, and EHD and mixed lubrication of different types of gears, not referenced.
- At Eq. 2 “(3)” to be deleted.
- Eq. (2): There are more sophisticated equations for the determination of the module.
- It is much better to start the calculation with the determination of the center distance based on the maximum tooth contact pressure.
- Eqs. (6) does not ensure the fulfilment of the condition of the same maximum specific sliding in both gears.
- In Eq. (9) “sen” to be changed to “sin”.
- In the ISO and DIN standards there are more sophisticated equations for the calculation on bending and pitting than Eqs. (11) and (12).
- The lubrication of gears is based only on the empirical results.
Author Response
Dear Editor, Dear Reviewer,
We thank you very much for the careful review of our article and for your recommendations for improvement. The paper has now been revised, and all the points raised have been properly tackled. The performed changes are clearly highlighted in yellow throughout the Manuscript file. In the following lines, a point-by-point response is addressed directly to the Reviewer (“>Q:” indicates the Reviewer’s Query being quoted, whereas “>>R:” introduces our reply).
REVIEWER 1:
>Q: The title to be changed to: “A Review About A Practical Approach to Cylindrical Gear Design and Lubrication”. The whole paper is only on the design of spur gears.
>>R: Considering the described design procedure may also be extended to cylindrical helical gears, which has been better highlighted in the paper text, the Manuscript title has been amended as recommended.
>Q: V. Simon have published a big number of papers on the design, manufacture, and EHD and mixed lubrication of different types of gears, not referenced.
>>R: A specific Reference with particular regard to EHD lubrication of gears has been incorporated into the Manuscript.
>Q: At Eq. 2 “(3)” to be deleted.
>>R: It has been checked and fixed.
>Q: Eq. (2): There are more sophisticated equations for the determination of the module.
>>R: The Authors agree with this remark: in fact, this equation is suitable for a preliminary rough estimation of the gear module at an early stage of the design process. More refined equations are provided in the following and make it possible to refine and optimize the module choice.
>Q: It is much better to start the calculation with the determination of the center distance based on the maximum tooth contact pressure.
>>R: A flow chart in the appendix has been added in order to better clarify this point. The theoretical center distance is an input parameter, the imposed center distance can be selected by the designer introducing also profile shift.
>Q: 6. Eqs. (6) does not ensure the fulfilment of the condition of the same maximum specific sliding in both gears.
>>R: The sentence has been corrected according to Niemann and Henriot books
>Q: 7. In Eq. (9) “sen” to be changed to “sin”.
>>R: It has been fixed.
>Q: In the ISO and DIN standards there are more sophisticated equations for the calculation on bending and pitting than Eqs. (11) and (12).
>>R: Yes, the more sofisticated formulas or equations according to the standards or to specific software (i.e. Kisssoft) can be used for the final verification, once the design is completed. We are focused on the design phase.
>Q: The lubrication of gears is based only on the empirical results.
>>R: Our aim was to provide a contribution having the capability of steering the designer through gear development, including its design, dimensioning and final drawing, as well as its most suitable lubrication. Guidelines for lubrication are based on the tools in the scientific and technical literature and on the features and specifications of the industrial application. The mentioned tools generally have an empirical source. More sophisticated numerical models are not proposed here, as they are beyond the scopes of the present study, also due to a lack of efficiency upon design stages.

Reviewer 2 Report
Sadly without a clear understanding of the novelty behind this paper, it was very difficult to understand and follow this manuscript. While there is evidence of some literature been included; particularly for section 3 of the manuscript, it is hard to understand why key tribological literature have not been considered and/or mentioned. It is felt the abstract, introduction and conclusion to be revised to incorporate the novelty of the paper. With the current state of the manuscript it is very difficult to recommend publication.
Major Issues:
- It is difficult to understand the major novelty behind the paper.
- As a general rule, the reader should be able to understand from the paper what is the new knowledge being contributed by this paper. These aspects must be clearly stated in the abstract, end of introduction and mentioned in the conclusion. The current manuscript does not appear to follow this rule.
- It is very difficult to distinguish between original ideas contributed by the authors and what is contributed by literature referenced in this manuscript.
- Following on from the earlier point; can the authors answer the following questions?
- Although EHL is mentioned in section 3, the authors do not mentioned the other forms of lubrication that gears can exhibit to. There should be justification why only EHL is being considered. Can the authors explain or justify how their study would only be in EHL and not mixed or boundary regime of lubrication?
- Based on the assumption this manuscript is supposed to give an overall overview to designing with appreciation for lubrication; there seemed to be a lack of the basic tribology literature. The following lists some of the available literature which would be essential for when choosing appropriate level surface roughness to avoid mixed/boundary friction.
- Dowson, D. and Higginson, G.R. Elastohydrodynamic Lubrication, Pergamon, Oxford (1977)
- Hamrock, B.J. and Jacobson, B.O., 1984. Elastohydrodynamic lubrication of line contacts. ASLE transactions, 27(4), pp.275-287.
- There is no mention of surface roughness. Would this not be an important aspect to consider? If so why have the authors not mentioned surface roughness and the above literature?
- Incidentally, the current manuscript implies the work is based on a type of spur gear. However, the authors haven’t specifically mentioned which gears the work/equations are applicable to.
- If the manuscript is supposed to be for a wide range of gears, this would mean elliptical contacts would also need to be considered.
- Following from point 2, the following literature would need to be also considered:
- Chittenden, R.J., Dowson, D., Dunn, J.F. and Taylor, C.M., 1985. A theoretical analysis of the isothermal elastohydrodynamic lubrication of concentrated contacts. I. Direction of lubricant entrainment coincident with the major axis of the Hertzian contact ellipse. Proceedings of the Royal Society of London. A. Mathematical and Physical Sciences, 397(1813), pp.245-269.
- Regarding section 3, the literature behind numerical approaches available for EHL contacts appears a little weak. It is advisable to show some general aspects of numerical modelling (such examples shown below)
- It would be interesting for the authors to make a case whether or not numerical approach is ideal for engineers (for designing purposes)
-
- Venner, C.H., 1992. Multilevel solution of the EHL line and point-contact problems.
- Almqvist, T. and Larsson, R., 2002. The Navier–Stokes approach for thermal EHL line contact solutions. Tribology International, 35(3), pp.163-170.
-
- It would be interesting for the authors to make a case whether or not numerical approach is ideal for engineers (for designing purposes)
Minor Issue
- Figure 4 seems to be very similar to the following reference. Can the authors confirm it is an original figure? Otherwise proper referencing is essential.
- Budynas, R.G. and Nisbett, K.J., 2014. Shigley's mechanical engineering design (in SI units).
- There is an absence of a dedicated nomenclature section
- It is advisable terms such as those listed from line 93 to 112 to be in the dedicated nomenclature.
- Hence, section 2 would need to be modified accordingly.
Author Response
Dear Editor, Dear Reviewer,
We thank you very much for the careful review of our article and for your recommendations for improvement. The paper has now been revised, and all the points raised have been properly tackled. The performed changes are clearly highlighted in yellow throughout the Manuscript file. In the following lines, a point-by-point response is addressed directly to the Reviewer (“>Q:” indicates the Reviewer’s Query being quoted, whereas “>>R:” introduces our reply).
REVIEWER 2:
>Q: particularly for section 3 of the manuscript, it is hard to understand why key tribological literature have not been considered and/or mentioned.
>>R: Section 3 has been amended and integrated with a detailed description of the three main regimes of friction and lubrication, i.e.: boundary friction, mixed friction and elasto-hydrodynamic (EHD) lubrication. The tribological conditions of the latter, being related to the tooth flank roughness conditions, have been highlighted as well. In addition, a further important reference with regard to elasto-hydrodynamic (EHD) lubrication has been incorporated into the Manuscript.
Q: > 1. It is difficult to understand the major novelty behind the paper.
As a general rule, the reader should be able to understand from the paper what is the new knowledge being contributed by this paper. These aspects must be clearly stated in the abstract, end of introduction and mentioned in the conclusion. The current manuscript does not appear to follow this rule.
It is very difficult to distinguish between original ideas contributed by the authors and what is contributed by literature referenced in this manuscript.
>>R: Complying with paper types in “Lubricants”, this is a Review paper that aims at steering the designer through gear development, based on industrial specifications, including, on one hand its design, dimensioning and final drawing, on the other hand, its most suitable lubrication. Regarding the first point, this paper deals with a collection of formulas tables and graphs taken from Standards and scientific and technical studies in the literature. These formulas have sometimes been revisited, to account for gear design (instead of just verification) purposes. As for the second point, guidelines for lubrication are also based on the available tools in the scientific and technical literature and are strictly related to the main features of the industrial application. Issues of novelty arise from the lack of scientific or technical review studies, addressing this topic and providing what van be regarded as a full-comprehensive collection of the aforementioned tools that are expected to significantly support the design task. As required, this passage has been included in the Introduction Section and, more concisely, in the Abstract and in the Conclusions.
>Q: Although EHL is mentioned in section 3, the authors do not mentioned the other forms of lubrication that gears can exhibit to. There should be justification why only EHL is being considered. Can the authors explain or justify how their study would only be in EHL and not mixed or boundary regime of lubrication?
>>R: As above, Section 3 has been amended and integrated with a detailed description of the three main regimes of friction and lubrication, i.e.: boundary friction, mixed friction and elasto-hydrodynamic (EHD) lubrication. The tribological conditions of the latter, being related to the tooth flank roughness conditions, have been highlighted as well. In addition, a further important reference with regard to elasto-hydrodynamic (EHD) lubrication, to be regarded as the most frequent, has been incorporated into the Manuscript.
>Q: Based on the assumption this manuscript is supposed to give an overall overview to designing with appreciation for lubrication; there seemed to be a lack of the basic tribology literature. The following lists some of the available literature which would be essential for when choosing appropriate level surface roughness to avoid mixed/boundary friction.
Dowson, D. and Higginson, G.R. Elastohydrodynamic Lubrication, Pergamon, Oxford (1977)
Hamrock, B.J. and Jacobson, B.O., 1984. Elastohydrodynamic lubrication of line contacts. ASLE transactions, 27(4), pp.275-287.
>>R: These studies have been included in the Reference list.
>Q: There is no mention of surface roughness. Would this not be an important aspect to consider? If so why have the authors not mentioned surface roughness and the above literature?
>>R: Please, see above regarding tribological conditions for EHD lubrication.
>Q: Incidentally, the current manuscript implies the work is based on a type of spur gear. However, the authors haven’t specifically mentioned which gears the work/equations are applicable to.
>>R: The described design procedure may also be extended to cylindrical helical gears. Therefore, this point has been clarified in the paper and the Manuscript title has also been amended, so that it currently makes reference to “cylindrical gears”.
>Q: If the manuscript is supposed to be for a wide range of gears, this would mean elliptical contacts would also need to be considered.
>>R: A refined study of contacts is beyond the scopes of this study and, in general, beyond the design task. Contact and pitting assessments are addressed based on Standard recommendations and on formulas that may be extended to helical gears, provided the helix angle is introduced. A part has been added to the manuscript.
>Q: Following from point 2, the following literature would need to be also considered:
Chittenden, R.J., Dowson, D., Dunn, J.F. and Taylor, C.M., 1985. A theoretical analysis of the isothermal elastohydrodynamic lubrication of concentrated contacts. I. Direction of lubricant entrainment coincident with the major axis of the Hertzian contact ellipse. Proceedings of the Royal Society of London. A. Mathematical and Physical Sciences, 397(1813), pp.245-269.
>>R: This document has been included in the Reference list.
Q>: Regarding section 3, the literature behind numerical approaches available for EHL contacts appears a little weak. It is advisable to show some general aspects of numerical modelling (such examples shown below)
It would be interesting for the authors to make a case whether or not numerical approach is ideal for engineers (for designing purposes)
Venner, C.H., 1992. Multilevel solution of the EHL line and point-contact problems.
Almqvist, T. and Larsson, R., 2002. The Navier–Stokes approach for thermal EHL line contact solutions. Tribology International, 35(3), pp.163-170.
R>>: These documents have been included in the Reference list. As above, sophisticated numerical models are not proposed here, as they are beyond the scopes of the present study, also due to a lack of efficiency upon design stages. Moreover, high computational expensiveness of numerical modelling upon design is also highlighted in these References.
>Q: Figure 4 seems to be very similar to the following reference. Can the authors confirm it is an original figure? Otherwise proper referencing is essential.
>>R: A suitable Reference has been added.
>Q: There is an absence of a dedicated nomenclature section.
It is advisable terms such as those listed from line 93 to 112 to be in the dedicated nomenclature.
Hence, section 2 would need to be modified accordingly.
>>R: The Nomenclature has been included in the revised version. Other changes have been done accordingly.

Reviewer 3 Report
Paper No.: Lubricants-846129
Title: A Review About A Practical Approach to Gear Design and Lubrication
This article presents a guide for design of gears that takes into account the right profile shifting, the selection of the proper lubricant, the quality grade and tolerances needed and backlash. The design methodology is illustrated with a practical example.
General Comments:
- The topic of the article is interesting and useful to the reader; however, the writing needs to be improved as it lacks clarity and organization. There are grammatical errors throughout the paper. Author should perform a general review of the entire paper.
- There should be a separate nomenclature section to define all the terms used in the paper.
- The introduction of the paper needs to be revised to add thorough literature survey and introduce different steps and variables in the gear design process. It would be helpful to understand the steps if a flow chart is added clearly showing the input/output variables at each step. The example in Figure 1 and calculation of Q using Eq. (1) seem to be arbitrary to include in the first paragraph of the introduction.
- The numerical example should be explained in more detail.
The reviewer believes that the manuscript can be recommended for publication after major revision addressing these concerns. Detailed comments are as follows.
- Comment #1 – Line 15-16
Please rewrite the sentence.
- Comment #2 – Line 24
Please replace “On the same time” with “At the same time”
- Comment #3 – Equation 1
Does Eq. 1 define a range for Q? If so, please use the correct convention for range. Make this correction for all the instances where authors want to define a range.
- Comment #4 – Line 34
Please correct the units of the Q. “3” should be a superscript.
- Comment #5 – Table 1
– “a” should a subscript.
- Comment #6 – Equation 2
Equation 2 is missing. Please correct the order of equations.
- Comment #7 – Line 152
Please check the condition for motion regularity. What is the meaning of that equation?
- Comment #8 – Equation 4
Please recheck the Eq. 4.
- Comment #9 – Line 261
Figure 6a is missing. The numbering of figure 7 and 8 seems to be wrong as there are two figure with caption Figure 8.
- Comment #10 – Table 9
Please explain the usage of table 9 in more detail.
Author Response
Dear Editor, Dear Reviewer,
We thank you very much for the careful review of our article and for your recommendations for improvement. The paper has now been revised, and all the points raised have been properly tackled. The performed changes are clearly highlighted in yellow throughout the Manuscript file. In the following lines, a point-by-point response is addressed directly to the Reviewer (“>Q:” indicates the Reviewer’s Query being quoted, whereas “>>R:” introduces our reply).
REVIEWER 3:
>Q: The topic of the article is interesting and useful to the reader; however, the writing needs to be improved as it lacks clarity and organization. There are grammatical errors throughout the paper. Author should perform a general review of the entire paper.
>>R: The paper has been carefully checked by the Authors, who have English language C1/C2 certification. Several amendments have been made.
>Q: There should be a separate nomenclature section to define all the terms used in the paper.
>>R: The Nomenclature has been included in the revised version. Other changes have been done accordingly.
>Q: The introduction of the paper needs to be revised to add thorough literature survey and introduce different steps and variables in the gear design process. It would be helpful to understand the steps if a flow chart is added clearly showing the input/output variables at each step. The example in Figure 1 and calculation of Q using Eq. (1) seem to be arbitrary to include in the first paragraph of the introduction.
>>R: The main steps of the design task are resumed in a flow chart that has been included in a final Appendix. However, in the Authors’ opinion, it was better to include citation to this flowchart in Section 2, specifically dealing with Gear Design. Anyway, the Introduction Section has also been enriched with a final paragraph dealing with the scopes of the current study and with the issues of novelty of the present Review paper. The applicative example has been left unchanged in the Introduction Section, as it is strictly related to the case study at the end of the Manuscript.
>Q: Comment #1 – Line 15-16
Please rewrite the sentence.
>>R: The sentence has been rewritten.
>Q: Comment #2 – Line 24
Please replace “On the same time” with “At the same time”
>>R: Fixed.
>Q: Comment #3 – Equation 1
>>R: Does Eq. 1 define a range for Q? If so, please use the correct convention for range. Make this correction for all the instances where authors want to define a range.
Range indications have been corrected throughout the paper.
>Q: Comment #4 – Line 34
Please correct the units of the Q. “3” should be a superscript.
>>R: Fixed
>Q: Comment #5 – Table 1
– “a” should a subscript.
>>R: Fixed
>Q: Comment #6 – Equation 2
Equation 2 is missing. Please correct the order of equations.
>>R: Fixed
>Q: Comment #7 – Line 152
Please check the condition for motion regularity. What is the meaning of that equation?
>>R: In order to warrant motion regularity, the (transverse) contact ratio must be εα >[1.25 - 1.4].
>Q: Comment #8 – Equation 4
Please recheck the Eq. 4.
>>R: Checked.
>Q: Comment #9 – Line 261
Figure 6a is missing. The numbering of figure 7 and 8 seems to be wrong as there are two figure with caption Figure 8.
>>R: Fixed
>Q: Comment #10 – Table 9
Please explain the usage of table 9 in more detail.
>>R: Sentence at lines 321-323 has been recast in order to clarify the usage of Tab. 9

Reviewer 4 Report
Lubricantas 29 iunie 2020
A Review About APractical Approach to Gear design and Lubrication,
Dario Croccolo,...
- The main objective of the manuscript under review was to present practical methods and techniques in order to design gears with higher load capacity, increased efficiency and longer fatigue life. In this respect the bending fatigue and especially rolling contact fatigue have been chosen as the failure criteria. Significant influences due to gears material and corresponding heat treatments, the right profile shifting, elasto-hydrodynamic lubrication (EHD)regime and tolerances needed to necessary backlash are presented.
- The manuscript contains the needed informations, equations and tables, to guide to an automatic computation.
- The influence of the elastic shaft deformation on pressure distribution in the new non-Hertzian contact needs at least some comments:
- Under real running conditions the meshing is altered by various deviations of the machine parts involved but also by elastic deformations which determine transmission errors and perturbations of pressures distributions along the flank, mainly at its end sides. The hertzian distribution of pressure (eq.(12)) and contact area are no longer hertzian. The pressures concentrations accelerate the rolling contact fatigue (RCF) and wear phenomena resulting shorter lives for the gears subjected to rolling contact.
- The Finite Element Method or Semi-Analytical Methods can be used to obtain the pressure distribution along the flank. The quarter space conditions have to be considered for the end sides.
- Some corrections as end relief (ISO 6336 2006 [15]) are used to attenuate the pressure peaks at the end of contact area.
Author Response
Dear Editor, Dear Reviewer,
We thank you very much for the careful review of our article and for your recommendations for improvement. The paper has now been revised, and all the points raised have been properly tackled. The performed changes are clearly highlighted in yellow throughout the Manuscript file. In the following lines, a point-by-point response is addressed directly to the Reviewer (“>Q:” indicates the Reviewer’s Query being quoted, whereas “>>R:” introduces our reply).
REVIEWER 4:
>Q: Under real running conditions the meshing is altered by various deviations of the machine parts involved but also by elastic deformations which determine transmission errors and perturbations of pressures distributions along the flank, mainly at its end sides. The hertzian distribution of pressure (eq.(12)) and contact area are no longer hertzian. The pressures concentrations accelerate the rolling contact fatigue (RCF) and wear phenomena resulting shorter lives for the gears subjected to rolling contact.
The Finite Element Method or Semi-Analytical Methods can be used to obtain the pressure distribution along the flank. The quarter space conditions have to be considered for the end sides.
Some corrections as end relief (ISO 6336 2006 [15]) are used to attenuate the pressure peaks at the end of contact area.
>>R: Sophisticated numerical models dealing with Hertzian / non Hertzian contact or shaft deformations are not proposed here, as they are beyond the scopes of the present study, also due to a lack of efficiency upon design stages. A high computational expensiveness of contact numerical modelling upon design is also highlighted by some References that have been included in the Reference list. Contact / pitting assessments have been run here, based on International Standard recommendations.

Round 2
Reviewer 1 Report
- Page 8, line 188: “…based on the suggestions of Ref. [26-28]:” What is the basis of these suggestions?
- “Guidelines for lubrication are based on the tools in the scientific and technical literature…”. The guidelines for lubrication are based only on the tools in the technical literature.
Author Response
Dear Editor, Dear Reviewer,
We thank you very much for the careful review of our article. The paper has now been revised, and the 2 points raised have been properly tackled. The performed changes are clearly highlighted in green throughout the Manuscript file.
- A sentence has been added according to Henriot, pag. 136.
- The sentence has been corrected according to the reviewer comment
Reviewer 2 Report
The authors have gone through and implemented the suggested comments. The explanations provided are satisfactory and the manuscript is now ready for publication.
Author Response
Dear Editor, Dear Reviewer,
We thank you very much for the careful review of our article.
Reviewer 3 Report
The authors have revised the manuscript to address the concerns and implemented the suggested changes. The revised manuscript is satisfactory and can be published.
Author Response

(The authors gave the same response as above.)
